# Epinephrine Auto-Injector Prescription and Use: A Retrospective Analysis and Clinical Risk Assessment of Adult Patients Sensitized to Lipid Transfer Protein

**DOI:** 10.3390/nu14132706

**Published:** 2022-06-29

**Authors:** Sara Urbani, Arianna Aruanno, Antonio Gasbarrini, Alessandro Buonomo, Rossana Moroni, Caterina Sarnari, Angela Rizzi, Eleonora Nucera

**Affiliations:** 1UOSD Allergologia e Immunologia Clinica, Dipartimento Scienze Mediche e Chirurgiche, Fondazione Policlinico Universitario A. Gemelli IRCCS, 00168 Rome, Italy; sara.urbani@hotmail.it (S.U.); arianna.aruanno@policlinicogemelli.it (A.A.); alessandro.buonomo@policlinicogemelli.it (A.B.); caterinasarnari@gmail.com (C.S.); eleonora.nucera@policlinicogemelli.it (E.N.); 2UOC Medicina Interna e Gastroenterologia, Dipartimento Scienze Mediche e Chirurgiche, Fondazione Policlinico Universitario A. Gemelli IRCCS, 00168 Rome, Italy; antonio.gasbarrini@policlinicogemelli.it; 3Medicina e Chirurgia Traslazionale, Università Cattolica del Sacro Cuore, 00168 Rome, Italy; 4Direzione Scientifica, Fondazione Policlinico Universitario A. Gemelli IRCCS, 00168 Rome, Italy; rossana.moroni@policlinicogemelli.it

**Keywords:** LTP allergy, anaphylaxis, epinephrine, food-allergy, panallergen, follow-up, management

## Abstract

Lipid transfer proteins (LTPs) are widely widespread plant food allergens which represents the main cause of food allergy in adults living in the Mediterranean basin. The purpose of this study was to investigate in LTP patients the actual use of prescribed epinephrine auto-injector and appropriateness of its prescription. In addition, we investigated in these patients: (1) occurrence of new food reaction in the following three years after to diagnosis; (2) need and number of access to emergency services; (3) presence of possible predictive factors to further food reactions. One-hundred sixty-five adult patients sensitized to LTPs have been included. During follow-up, we recorded 68 further reactions, most of them (77.9%) characterized by local symptoms; rarely the patients required an emergency-department visits (16.1%) and only one patient (1.7%) used the epinephrine auto-injector. The patients with a previous history of anaphylaxis at baseline turned back to access to emergency services also during the follow-up (*p* = 0.006). The majority of patients with recorded systemic reactions (*p* = 0.004) and treated in an emergency room (*p* = 0.028) did not have any co-factor-enhanced at diagnosis. We noted an association between platanus pollen sensitization and severity of further reactions during the follow-up (*p* = 0.026). Epinephrine auto-injector were prescribed to 108/165 patients (65.5%) with an over-prescription rate of 25%. The unforeseeable clinical presentation of LTP allergic reactions and the eventual role played by the cofactor make necessary schedule a follow-up to monitor the patients over time and to assess the actual use of epinephrine auto injectors prescribed.

## 1. Introduction

Lipid transfer proteins (LTPs) are allergens widespread in plant foods and pollen belonging to the pathogenesis-related-14 (PR-14) family. LTPs perform antimicrobial and defense activities and for this reason they are highly conserved and expressed in the peel of the Rosaceae fruits and in a large number of vegetables [1,2].

LTPs are 7–9 kDa proteins, rich in cysteine residues, folded through sulfur bridges into a very compact structure, which are extremely heat stable and resistant to gastric proteolysis and food processing. This molecular feature increases systemic absorption and the occurrence of severe systemic allergic reactions [2].

Indeed, LTP represent the main cause of food allergy in adults living in the Mediterranean basin [3,4] and they are the main cause of primary food allergy in Italian adults with a prevalence of 9.8% [5]; less epidemiologic data are available for pediatric population [6,7]. Due to high cross-reactivity, they frequently cause clinically relevant allergic reactions to various plant foods in sensitized patients. LTP sensitization may occur through oral, inhaled or cutaneous route, and may be elicited by a pollen nsLTP (non-specific lipid transfer protein) [8]. The manifestation and severity of LTP hypersensitivity are extremely variable. Many patients are sensitized but completely asymptomatic, others may show exclusively local reactions, as contact urticaria or oral allergic syndrome (OAS), and others may present severe systemic symptoms up to anaphylaxis [9].

To complicate the clinical presentation, LTP-hypersensitive patient may show a spectrum of co-recognitions ranging from one single vegetable food to a large number of biotically unrelated foods [10]. Previous findings showed that this phenomenon is linked to the level of Pru p 3-specific IgE: higher Pru p 3 IgE levels are higher is the number of sensitized an co-recognized foods [11].

Moreover, LTP reactions can be triggered or exacerbated by cofactor such as nonsteroidal anti-inflammatory drugs (NSAIDs) administration, alcohol intake or physical exercise [12,13,14]. Given the peculiarities of this allergy and the unpredictability of its clinical manifestations, epinephrine auto-injector prescription becomes an important medico-legal and ethical issue.

To date, there is an indication to prescribe self-eligible epinephrine in patients who had already experienced severe systemic reactions. The clinical-therapeutic allergist behavior towards patients with local reactions or asymptomatically sensitized to LTP remain unresolved. Instead, few data are available about long-term follow-up of LTP sensitized patients and about onset of further food reactions to previously tolerated food.

## 2. Materials and Methods

### 2.1. Patients

The present study was carried out on 165 adult subjects with a diagnosis of LTP allergy from at least three years, presenting at the outpatient Allergy Unit of Fondazione Policlinico Universitario A. Gemelli IRCCS of Rome between 1 January 2018 and 31 December 2020. All enrolled participants had a diagnosis of LTP IgE-mediated allergy detected by skin prick test (SPT) and confirmed by specific rPru p3 IgE assay.

The Ethical Committee of our Institute evaluated and approved this study (ID 3742). An informed consent form was signed by all eligible and willing patients in accordance with the Declaration of Helsinki.

### 2.2. Study Design

In this observational retrospective single-centre study, we analyzed data collected by medical records. According to baseline visit (T0), patients were retrospectively classified into three different groups depending on reported adverse reaction to LTP foods: (1) patient with a clinical history of systemic reactions (urticaria/angioedema, anaphylaxis); (2) patients with a clinical history of local reactions (contact urticaria, oral allergic syndrome); (3) patients with clinical asymptomatic LTP sensitization. All patients were recommended to avoid culprit food(s) and to continue eating all other tolerated LTP food without association with known cofactors (exercise, alcohol and anti-inflammatory drugs). We informed patients about the correct therapeutic management of further reactions. Moreover, for three following years, we planned an annual visit with the patients to evaluate their clinical condition, eventual further reactions and to prescribe epinephrine auto-injector where necessary. In the aftermath, we evaluated the appropriated epinephrine prescriptions rate according to EAACI guidelines [15]. 

Considering the three following years from the diagnosis, we investigated: (1) the occurrence of further foods reaction; (2) the actual use of the prescribed epinephrine injector and the appropriateness of the epinephrine prescription; (3) need and number of access to emergency services or alternately corticosteroids and antihistamine therapy administration in home environment for signs and/or symptoms of food allergy.

Moreover, we aimed to recognize at the diagnosis the presence of possible predictive/predisposing clinical or laboratory data to further new food reactions (LTP basophils activation test levels, total levels of serum IgE, allergic comorbidities, co-sensitization to other panallergens). 

Lastly, we evaluated eventual “drop-out” of patient during the three years of follow-up and the association with their clinical presentation allergy.Afterward we conducted a telephonic interview to investigate the reasons for follow-up drop-out, classifying them into: (1) personal issues; (2) logistic reasons (distance from hospital, difficult to make an appointment); (3) poor perception of allergy severity.

### 2.3. In-Vivo and In-Vitro Test

Patients underwent an allergological work-up included skin prick tests (SPTs) with commercial extracts peach LTP, PR-10, profiline and other food and airborne allergens (Lofarma, Milan; Alk-Abellò, Milan, Italy). They were considered positive when a wheal at least 3 mm greater in diameter than the negative control was observed. Previous studies showed that commercial peach extract a (Alk-Abellò, Milan, Italy) virtually contains LTP at a concentration of 30 µg/mL and a positive SPT with this extract should be considered a clinical marker of sensitization to this protein. Furthermore, we confirmed SPTs results through total and specific serum assay (Thermo Fisher Scientific, Waltham, MA, USA) considering positive specific IgE values more than 0.35 kU/L. We measured serum specific IgE levels to rPru p 1, rPru p 3, rPru p 4, Pla a3, Par j 2.

### 2.4. Statistical Analysis 

The sample has been described in its demographic and clinical characteristics applying descriptive statistics techniques. Categorical variables have been presented as absolute frequencies and percentages. Quantitative variables have been summarized with mean and standard deviations. The normality of data has been verified with the Kolmogorov-Smirnov test. 

Associations between categorical variables have been evaluated with the Chi-square test (or Fisher’s Exact Test when required). A logistic regression was performed for the clinical variables with dichotomous scores to investigate whether associations between clinical characteristics and patient loss of follow-up were present. A *p*-value < 0.05 was considered statistically significant. All the statistical analyses have been performed with SPSS 25 (IBM Corp, Armonk, NY).

## 3. Results

### 3.1. Patients at Baseline 

A total of 165 patients have been included in the study, 110 females (66.7%) and 55 males (33.3%) with a mean age of 37.8 ± 12.4 years. Main demographical and clinical characteristics of patients are summarized in Table 1.

After LTP food ingestion (T0), 88 patients (53.3%) had a clinical history of local reaction, 54 (32.7%) reported a systemic reaction and particularly 13 (7.9%) patients recorded a history of food-induced anaphylaxis (according to WAO diagnostic criteria [16]). The remaining 10 (6.1%) patients had a LTP asymptomatic sensitization. Concerning the number of reactions that happened before the diagnosis, 51 patients (30.9 %) had more than 5 reactions to different types of food, 40 patients (24.2%) experienced more than 3 previous reactions, whereas 57 patients (34.5%) had only 1–2 reactions. 57 patients (33.9%) presented reactions in association with a cofactor, in details 11 (6.5%) with alcohol, 17 (10.1%) with physical exercise and 29 (42.6%) with a concomitant non-steroidal anti-inflammatory drugs (NSAIDS) administration. In 88.5% of cases the reactions were characterized by cutaneous symptoms, in 19% by respiratory distress and in the 23% gastrointestinal symptoms. The same patient could have experienced more than one kind of reaction. For the management of food severe adverse reactions, 41.2% of cases were evaluated by emergency department visit; 65 patients (39.4%) required the use of parenteral corticosteroids and antihistamines and 8 patients (4.8%) required epinephrine administration. 

### 3.2. Epinephrine Prescription and Actual Use 

Epinephrine auto-injector were prescribed to 108/165 patients (65.5%) in face of 67/165 (40.6%) patients that have a strict indication. Indeed, based only on the clinical history, we recorded an epinephrine over-prescription of 25%. To establish the epinephrine appropriateness prescription, we based on EAACI guideline [15] that provide as absolute indications for epinephrine auto injectors prescription in case of: (a) previous anaphylaxis triggered by food (*n* = 13, 12%) previous exercise-induced anaphylaxis (*n* = 5; 4.5%), co-existing unstable moderate-severe-persistent asthma (*n* = 1, 0.9%). Moreover, we considered appropriate the prescription also in presence of following additional factors: (a) Severe systemic allergic reactions (*n* = 42, 38.8 %), (b) mild-to-moderate reactions in teenager or young adult (*n* = 7, 6.4%) (c) considerable distance from medical help or prolonged travel abroad (*n* = 4,3.7%). Considering that previously mentioned conditions could be coexisting in the same patients, we estimate that 67 patients had a strict epinephrine auto-injectors prescription indication. The 41 remaining patients presented at least one moderate-to-severe signs or symptoms of reactions related to LTP containing food, previously evaluated by emergency department (Figure 1). 

Compared to a high rate of epinephrine prescriptions, we recorded a low rate of use by the allergic patients. During an anaphylactic reaction, only a patient (1.7%) needed to use epinephrine auto-injector waiting for rescue.

### 3.3. Further Reactions Recorded during the Follow-Up

We monitored the onset of further reactions during the next three years of follow-up (Table 2) recording 68 new reactions during the follow-up with a rate of further reactions of 41% (68/165) considering total patients and 60% (68/108) considering only the patients remaining in the follow-up.

In particular, we noted 33.8% (23/68) of further reactions during the first year, 41.2% (28/68) during the second year and 25% (17/68) during the third one. Most reactions (53/68, 77.9%) were characterized by local symptoms that promptly receded with home therapy (oral antihistamines and corticosteroid); we recorded systemic symptoms in 19.1% (13/68) cases and 2.9% (2/68) of anaphylaxis. Rarely the patients required an emergency-department visit (16.1%) and only one patient (1.7%) used the epinephrine auto-injector waiting for rescue. Among the patients with an asymptomatic sensitization to LTP at diagnosis, we recorded three new reactions, one of them characterized by severe systemic symptoms. The patients with previous history of anaphylaxis treated in Emergency Room at T0 turned back to emergency services also during the follow-up (*p* = 0.006). The majority of patients with recorded systemic reactions (*p* = 0.004) and the majority of patients treated in an Emergency Room (*p* = 0.028) did not have of co-factor-enhanced food allergy history at diagnosis. 

Moreover, we noted an association between platanus pollen sensitization (Pla a 3) and severity of further reactions during the follow-up (*p* = 0.026). No clinical correlations were observed between T0 clinical history (asymptomatic, local reaction, systemic reactions) and age, sex (*p* = 0.397), sensitization to other panallergens (*p* = 0.436), sensitization to platanus (*p* = 0.945) or parietaria (*p* = 0.656) pollens, further reactions during follow-up (*p* = 0.704), rPrup 3/ rPar j 2/ Pla a 3 specific IgE values (*p* > 0.05), total serum IgE/ LTP basophils activation test (both *p* > 0.05), cofactors involved (*p* = 0.087). Significant correlations between clinical factors presented at the diagnosis and number/severity of further reactions during the follow-up are summarized in Figure 2.

### 3.4. Adherence to Follow-Up

Overall, 108 patients (65.5%) were lost during the follow-up; particularly 60 (35.7%) at first year, 32 (19%) at second year and 16 (9.5%) at the third one. 61.1% of lost patients were female and 38.9% were male (*p* = 0.037). A large proportion of patients (86%) who attended the annual follow-up had epinephrine auto-injector prescription. Logistic regression analysis revealed that the patients without epinephrine auto-injector prescription were lost 5 times more than the patients with it (*p* < 0.0005, HR = 5.08, 95%CI =2.2–11.7). A higher proportion of patients without a concomitant sensitization to other panallergens (profiline, PR-10) left the follow-up (80.2% vs 19.8%, *p* = 0.042, HR = 2.211, 95%CI = 1.031–4.740). On the contrary, the presence of other allergic comorbidity (allergic rhinitis, asthma, other food allergy, drug allergy, etc.) did not influence the adherence to the follow up (*p* = 0.331) as well as the number of adverse food reactions (*p* = 0.601) and the number of foods involved in (*p* = 0.135) before the diagnosis. 

Furthermore, a telephone interview was conducted to check the drop out reasons. 38 (35.1%) of patient have not returned to medical attention due to personal issues, 12 (11.1%) to logistic issues and 49 (45.3%) to poor perception of their allergy severity; 9 patients (8.3%) were out of reach. 

## 4. Discussion

The unpredictable clinical presentation of LTPs allergy and the possibility of development of new reactions towards previous tolerated food complicate long-term management of patients and the decision of epinephrine prescriptions by the clinicians. 

To our knowledge, this is the first study designed to investigate the actual use of epinephrine auto-injector by LTP allergic patients. A few data are currently available about the sensitization profile of patients and about the risk of new reactions in the years following the diagnosis [9,17]. Compared to previous reports, we observed 41% (68/165) of further reactions during the follow up versus 27% (18/67) of Asero et al. [9] and 31% (35/113) of Betancor et al. [17]. Similarly, to previous findings [9], most of the reaction (78%) consisted in local symptoms easily treated with oral antihistamines and/or oral corticosteroids administered in home environment. The severity of reactions during the follow-up correlated with: (1) absence of cofactors (*p* = 0.028) demonstrating that in these cases the food alone was sufficient to elicit the reaction; (2) previous systemic reactions (*p* = 0.004) and Emergency Room access (*p* = 0.028); (3) platanus sensitizations, as already demonstrated by previous work [18].

In our Centre, we recorded an epinephrine auto-injector over-prescription rate of 25%. A similar excess of epinephrine prescription (equal to 29%) in LTP allergic patients have also been reported from Asero et al. [19]. The over prescription of epinephrine autoinjectors may be related to the peculiarity of the LTP allergy, which is not associated to a single allergen, but to a family of highly cross-reactive proteins. Hence, food-hidden allergens and sensitizations to previous tolerated food represent a concrete possibility in everyday patients’ life [8]. Moreover, the severity of one allergic reaction does not able to predict the severity of future ones [20]; indeed, the patient who experienced a mild-moderate reactions can have a life-threatening or lethal reaction with a subsequent exposure [21]. 

In this scenario, with multiple studies demonstrating that mortality from anaphylaxis is associated with delay or lack of epinephrine use [22,23,24], clinicians may over prescribe auto-injectors considering their life-saving role in case of severe reactions. Our over-prescription data could be interpreted as a clinical need to protect the patient to future reactions, considering that LTP is the most important allergen causing food-induced anaphylaxis in Italy [4]. 

However, in face of epinephrine auto-injection over-rate prescription in our LTP patients we documented a low rate of epinephrine auto-injector use (1.7%) and emergency department access (6.6%) during the observation period. It should be noted that the data on the real use of adrenaline are available only for subjects who have returned annually to medical attention. These patients were adequately informed from year to year on the management of allergic reactions and provided with written, personalized emergency action plans.

For this reason, these data are straitly influenced to an adequate patient education in refraining culprit food ingestion or highly cross-reactive food and in respecting a recommended behavior (e.g., peeling fruits, avoiding the cofactors associations, regular ingestion of tolerated food). As an alternative to indiscriminate epinephrine prescription the clinicians should be considered the crucial role of patients’ and their caregivers’ education in preventing anaphylaxis since first evaluation of patient [24]. Indeed, considerable loss to follow up, suggest education should start early in order to increase patients’ perception on allergy severity and improve their adherence to the follow-up.

Therefore, the unforeseeable clinical presentation of further reactions and the eventual role played by the cofactor make necessary to schedule a follow-up to monitor the patients over the time, giving them all materials and tools to management of reaction (including epinephrine autoinjector when necessary). Furthermore, epinephrine auto-injector prescription seems to be increasing patient awareness of severity of their allergy and their adherence to the follow-up. Indeed, our results show that the patients with an epinephrine auto-injector prescription were lost 5 times less than the patients without it (*p* < 0.0005). On the other hand, the excessive prescription of epinephrine auto-injector does not seem to be associated with an inadequate use of it. 

Considering that in our study we collected a real-life retrospective data, its main limitation consists in the high rate of loss to follow-up over the 3 years period. The comparatively high level of loss to follow-up is potentially attributable to several reason: a perceived allergological low risk by patients, a low rate of further adverse reaction occurred among these patients or alternatively among a perceived lack of utility of further follow up. Nevertheless, calling back the patients lost to follow-up we often found a poor perception of risks related to their allergy. For these reasons the first communication with patients should be focused on possible risks of further sensitization to LTPs food and on correlations between cofactors and severe reactions.

Overall, physician-patient communication is an essential element of care; patients with a LTP allergy diagnosis should be counseled on avoidance measurement (e.g., reading food labels carefully, abstaining from cross-reactivity food ingestion, no cofactor associations) either in the first evaluation or in the next ones. Frequent monitoring and annual medical advice should be improved in patient affected by LTP allergy to increase their awareness, confidence and expertise in allergic reaction recognition and management and in the correct use of epinephrine autoinjector. Visual aids, such as video, simple treatment algorithm or use of simulation anaphylaxis scenario can be a helpful reminder for the patients in learning the correct behavior in front of reactions and in the therapeutic decision making [24]. 

## 5. Conclusions

In summary, the actual use of epinephrine autoinjector as well as incidence of severe allergic reactions seem be low in LTP patients previous diagnosed and taken charge by allergy unit. The relative epinephrine over-prescription rate could be justified by the peculiar and unpredictable features of this allergy. Further investigations are useful to phenotype these patients defining their risk profile and consequent their real epinephrine autoinjector need, optimizing individual treatment.

## Figures and Tables

**Figure 1 nutrients-14-02706-f001:**
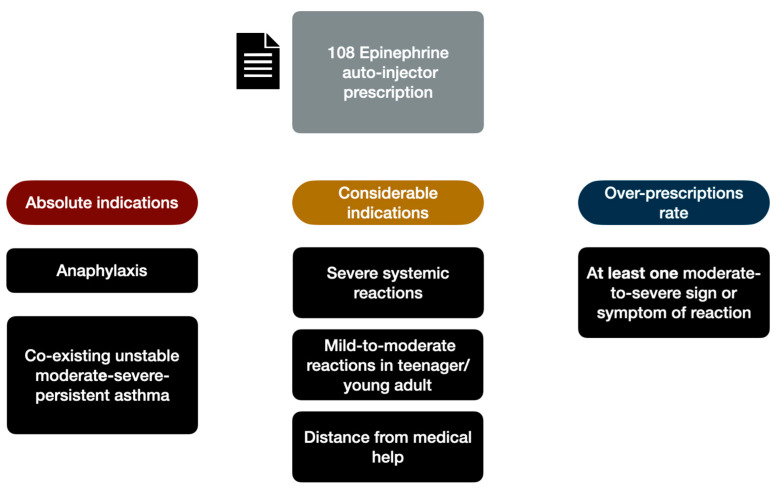
Epinephrine appropriateness prescription based on EAACI guideline [15].

**Figure 2 nutrients-14-02706-f002:**
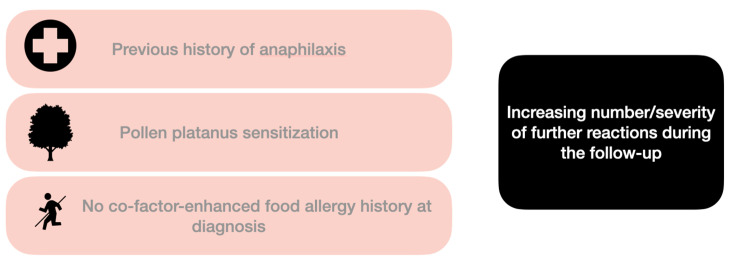
Main associations between clinical factors at the diagnosis and number/severity of further reactions during the follow-up. An association (assessed by Chi-square test) is noted with a previous history of anaphylaxis treated in Emergency Room at T0 (*p* = 0.006), absence of co-factor-enhanced at diagnosis (*p* = 0.004), platanus pollen sensitization (*p* = 0.026).

**Table 1 nutrients-14-02706-t001:** Demographic and clinical characteristics of patients (T0)

Characteristics	Values
Female, *n* (%)	110 (66.7)
Age (years)	37.8 ± 12.4
BMI (kg/m^2^)	23.4 ± 3.9
Smoking, *n* (%)	22 (13.3)
Allergy family history	59 (35.8)
**Comorbidity**	
Allergic rhinitis (%)	102 (61.8)
Contact dermatitis, *n* (%)	18 (10.9)
Atopic dermatitis, *n* (%)	14 (8.5)
Drug allergy, *n* (%)	13 (7.9)
Latex allergy, *n* (%)	5 (3%)
Asthma, *n* (%)	1 (0.6)
**Clinical history (T0)**	
Asymptomatic, *n* (%)	10 (6.1)
Local reactions	88 (53.3)
Systemic reactions	54 (32.7)
Anaphylaxis	13 (7.9)
**Number of reactions before diagnosis**	
0, *n* (%)	12 (7.3)
1 or 2, *n* (%)	57 (34.5)
>3, *n* (%)	96 (58.1)
**Clinical characteristics of reactions (T0) ***	
Cutaneous symptoms	146 (88.5)
Respiratory symptoms	32 (19.4)
Gastrointestinal symptoms	38 (23)
**Therapy (T0)**	
Spontaneous resolution	41 (24.8)
Oral anti-H1/CCS	48 (29.1)
Parenteral anti-H1/CCS	65 (39.4)
Parenteral anti-H1/CCS	8 (4.8)
Missing	3 (18)
**Epinephrine prescription**	108 (65.5)

Data are presented as mean ± SD or %, as indicated. BMI = body mass index; CCS = corticosteroids, anti-H1 = antihistamines. * The sum is more than 100% as the same patient experienced more than one type of reaction.

**Table 2 nutrients-14-02706-t002:** Clinical severity and therapy of further reactions recorded during the follow-up.

		1st Year	2nd Year	3rd Year
Remaining patients at the beginning of the period		165	105	73
Lost patients *n*, (%)		60, (36.4%)	32, (19.4%)	16, (9.5%)
Further reactions *n*, (%)	local symptoms *n*, (%)	16, (70%)	16, (70%)	22, (79%)
	systemic reaction *n*, (%)	6, (26%)	6, (26%)	5, (18%)
	anaphylaxis, (%)	1, (4%)	1, (4%)	1, (3%)
	home-therapy, (%)	16 (79.9%)	16 (79%)	18, (82%)
Therapy	emergency department visit, (%)	4, (17%)	4, (17%)	6, (27%)
	use of epinephrine auto-injector, (%)	0	0	1, (4%)

Percentages calculated on the basis of patients remaining in the follow-up.

## Data Availability

Data sharing is not applicable to this article.

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
