# Peer review of "Epinephrine Auto-Injector Prescription and Use: A Retrospective Analysis and Clinical Risk Assessment of Adult Patients Sensitized to Lipid Transfer Protein"

_nutrients, 2022, doi:10.3390/nu14132706_

Round 1

Author Response

To Guest Editor and Reviewers
Nutrients MDPI

We would like to greatly thank the Editor and Reviewers who encouraged a complete revision of the manuscript.

Please find the enclosed the Revision vers. 1 of the Review article entitled “Epinephrine auto-injector prescription and use: a retrospective analysis and clinical risk assessment of adult patients sensitized to lipid transfer protein” by Sara Urbani, Arianna Aruanno, Antonio Gasbarrini, Alessandro Buonomo, Rossana Moroni, Caterina Sarnari, Angela Rizzi and Eleonora Nucera.

[Nutrients] Manuscript ID: nutrients-1705670 - Major Revisions

Author's Reply to the Review Report (Reviewer 1)
Comments and Suggestions for Authors

Overall
As antimicrobial and defense proteins, LTPs are highly conserved panallergens, present in a wide variety of plant foods like Rosaceae fruits and many vegetables as well as in pollens.
There is a consensual indication to prescribe self-eligible epinephrine in patients who had already experienced severe systemic allergic reactions. However, possibly because of reported severe reactions associated to food-hidden allergens, doctors may be over-prescribing epinephrine auto-injectors. This observational retrospective single-centre study analyzed data collected from medical records. A total of 165 adult subjects with a diagnosis of LTP allergy were studied in terms of clinical evidence of real need for epinephrine prescription and performed a discussion.

Abstract

  1. “(2) actual use of prescribed epinephrine auto-injector and need…” How do the authors identify cases of prescription with no need? Isn’t it associated with “(4) appropriateness of the epinephrine prescription”? If yes it would probably be better to include in the same point.
    We thank the Reviewer for the comment. We investigated the actual use of prescribed epinephrine autoinjector by the patient and farther/alternatively the need and the number of patients accesses to first aid centres to manage their allergic reactions. We modified the manuscript according to Reviewer’s comments.
  1. “Epinephrine auto-injector were prescribed to 108/165 patients (65.5%) with an over prescription rate of 25%.” It was not found clear the author’s rational for the need/no need of epinephrine auto-injector prescription. Was it a) the occurrence of a recorded systemic reaction with the need for treatment in an emergency room; b) the existence of a co-factor enhancing the diagnosis or c) both conditions? In the introduction section (lines 60-61) the authors refer that “…there is an indication to prescribe self-eligible epinephrine in patients who had already experienced severe systemic reactions.” It would result clearer if at least a short reference was made in the abstract to the current consensual rational of prescription.
    We thank the Reviewer for the comment. We added a section in the “Results” as suggested by Reviewer.
  1. The last sentence (lines 31-34) would also benefit of a rewriting for easier understanding.
    We thank the Reviewer for the comment. We modified accordingly.

Materials and Methods

  1. Lines 70-71 – The authors refer that “All enrolled patients had a diagnosis of LTP IgE-mediated allergy detected by skin prick test (SPT) and confirmed by specific rPru p3 IgE assay.” However, there were just sensitized asymptomatic individuals. Please adjust the text to that phenotype condition.
    We thank the Reviewer for the comment. We enrolled also sensitized asymptomatic individuals dividing the population in: “patient with a clinical history of systemic reactions (urticaria/angioedema, anaphylaxis); (2) patients with a clinical history of local reactions (contact urticaria, oral allergic syndrome); (3) patients with clinical asymptomatic LTP sensitization”. We decided to include them in order to observe also an eventual onset of clinical manifestation in this patient category (a few data are available in literature for long term clinical development of these patients).” Among the patients with an asymptomatic sensitization to LTP at diagnosis, we recorded three new reactions, one of them characterized by severe systemic symptoms”. We modified accordingly.
  1. Lines 94-97 – The authors refer that “we aimed to recognize at the diagnosis the presence of possible predictive/predisposing clinical or laboratory data to further new food reactions (LTP basophils activation test levels, total levels of serum IgE, allergic comorbidities, co-sensitization to other panallergens).”. Could the authors present any results of BAT (or regarding oral provocation tests if reported)?
    We thank the Reviewer for the comment. In some cases, BAT was performed and confirmed the diagnosis, but we did not find a statistical correlation between its values and the clinical presentation of patients. The oral provocation test was performed only for the reintroduction of those food that were doubtfully involved in adverse reactions. We modified accordingly.

Results and Discussion
Table 1.

  1. Irregular space between lines (probably an editing issue). Please attend to that.
    We thank the Reviewer for the comment. We modified accordingly.
  1. When the authors present data such as “Systemic reactions”, “Anaphylaxis” and “Epinephrine administration” do they mean at all or only related to suspected allergy to LTPs (information from clinical records)? It is relevant as despite the aim to recognize possible predictive laboratory test results from LTP basophil activation, no controlled provocation tests were mentioned or probably performed. Please clarify.
    We thank the Reviewer for the comment. The diagnosis of LTP allergy was based on the combination of clinical history and allergological diagnostic work-up. Since the retrospective nature of this study, BAT and oral provocation test did not performed systematically for each patients. We modified accordingly.
  1. Line 108 – At the time the work was performed and patients enrolled the company was already “Thermo Fisher Scientific, Waltham, MA, USA“. Please correct if considered appropriate.
    We thank the Reviewer for the comment. We modified accordingly.
  1. Lines 141-146 – The authors refer that “For the management of food severe adverse reactions, 41.2% of cases were evaluated by emergency department visit; 65 patients (39.4%) required the use of parenteral corticosteroids and antihistamines and 8 patients (4.8%) required epinephrine administration. Epinephrine auto-injector was prescribed to 108/165 patients (65.5%) in face of 67/165 (40.6%) patients that experienced a severe systemic reaction. Indeed, based only on the clinical history, we recorded an epinephrine over prescription of 25%.” These observations are impressive and lead to an immediate conclusion of over-prescription. However, as seen from figures in Table 1, most patients presented with or referred 1-2 (34.5%) or >3 (58.1%) reactions before diagnosis. Did the authors check for any information regarding possible aggravation of clinical signs through the successive episodes and could it be associated to the preventative prescription of epinephrine auto-injector? Nevertheless, the reported “…high rate of loss to follow-up over the 3 years period.” (Line 264) stands as a matter of concern when prescribing epinephrine for self-medication. Even though “…patients with an epinephrine auto-injector prescription were lost 5 times less than the patients without it…” (lines 285-286) and it may reveal association with further susceptibility to severe reactions. This dual issue – possible emergency need vs loss of follow-up would benefit from discussion when addressing the rational of decision.
    We thank the Reviewer for the comment. We modified accordingly.
  1. As epinephrin auto-injector may be considered as a life-line guardian to use when a severe condition happens (similar to a defibrillator in a public place) and food-hidden allergens are a real possibility, could that prescription be considered not so excessive, after all? In fact, the first sentence of the discussion shows a possible reflex of that.
    We thank the Reviewer for the comment. This work aimed to demonstrate that in LTP allergy the epinephrine prescription could not be guided solely on clinical previous history of anaphylaxis or presence of cofactor-enhanced. Clinicians should consider the LTP allergy peculiarities not only linked to a single allergen, but a family of proteins highly cross-reactive each other. Considering this allergen features, the prescriptions not appeared too excessive. We modified accordingly.
  1. Lines 283-284 – Sentence should be rewritten.
    We thank the Reviewer for the comment. We modified accordingly.

Best regards,

Sara Urbani, Arianna Aruanno, Antonio Gasbarrini, Alessandro Buonomo, Rossana Moroni, Caterina Sarnari, Angela Rizzi and Eleonora Nucera

Reviewer 2 Report

Thank you for the opportunity to review the manuscript, Epinephrine auto-injector prescription and use: a retrospective analysis and clinical risk assessment of adult patients sensitized to lipid transfer protein. In this manuscript, the authors made use of retrospective data from 165 adults sensitized to LTP, amongst whom there were 68 reactions (predominantly minor/localized symptoms), of which only one required administration of epinephrine. Although an interesting and timely manuscript given an increasing focus on LTP, there are many concerns with the manuscript in its current form.

  1. Both the abstract and the main text lack an explicit aim.
  2. In Line 20, the authors report that they investigated “further food reaction” over 3 years. Further to what? As this is the first mention of food reaction related to their study population, it is unclear to what these reactions are further.
  3. Line 21: First aid centres, in English, typically refer to small clinics where first aid for minor injuries is provided. Do the authors mean emergency services?
  4. The abstract lacks a definition of “appropriateness of the epinephrine prescription.” This needs to be operationalized, given that it is one of the primary outcomes.
  5. What types of statistical analyses were performed? One wonders if the authors performed associations (more correctly) than correlations, although “correlation” is mentioned in Line 29.
  6. Please clarify what is mean by “pollen platanus”
  7. In Lines 21-22, the authors refer to “possible predictive factors.” Is this the same as the “cofactor” described in Line 32?
  8. Line 39: “Pan allergens widespread” is redundant. Reword to “allergens widespread”
  9. Line 40 “carry out” is colloquial. Please revise.
  10. Lines 45-46: “This molecular feature increases…. the occurrence of severe systemic reactions.” Perhaps the authors mean “increases the risk of severe systemic reactions”?
  11. Section 2.1: This is more correctly “participants” than “patients”
  12. Line 85: What do the authors mean by “we programmed to patient an annual visit…”
  13. In the paragraph from Lines 76-93, the authors write about the appropriateness of a prescription for epinephrine autoinjectors. Yet, there is neither an operationalized definition for appropriateness, nor a discussion as to how appropriateness was determined (who wrote the initial prescription, who determined appropriateness and based on what criteria, etc). Without this information, the reader cannot appropriately interpret these findings.
  14. With consideration to the reactions reported, were data collected on the form of the food that triggered the reaction (i.e. cooked, vs raw)?
  15. Line 113: Rather than qualitative variables, do the authors mean “descriptive” variables? Qualitative data refer to interviews, observations, photographs and other types of data reported by participants to glean insight into lived experience.
  16. Line 113: Rather than quantitative variables, do the authors mean “continuous” variables?
  17. In some of the results, observations are less than 5. Rather than a chi-square test, it would be appropriate to use a Fisher Exact Test.
  18. The results section is very long. Please consider including subheadings that align with the outcomes of interest to help guide the reader.
  19. Lines 166-7: The loss to follow up is substantial, considering a relatively small sample size. Please comment, and/or consider multiple imputation to maximize the data.
  20. Table 2: A small comment: The degree (superscript o) after the number of each year is not used in English.
  21. Table 2: A small comment: What do the commas mean following some of the results?
  22. In Line 212, the authors mention “logistic regression.” This was a bit of a surprise, given that there was no mention of this type of test in the section describing statistical analysis methods.
  23. Lines 245-7: I suspect that these lines were included in the original template. In the submission, they should have been removed.
  24. Although one of the outcomes of interest was co-factors, it is unclear how these data were collected. This needs to be described in the methods.
  25. The manuscript lacks a conclusion that aligns with the aim (also missing).

Author Response

To Guest Editor and Reviewers
Nutrients MDPI

We would like to greatly thank the Editor and Reviewers who encouraged a complete revision of the manuscript.

Please find the enclosed the Revision vers. 1 of the Review article entitled “Epinephrine auto-injector prescription and use: a retrospective analysis and clinical risk assessment of adult patients sensitized to lipid transfer protein” by Sara Urbani, Arianna Aruanno, Antonio Gasbarrini, Alessandro Buonomo, Rossana Moroni, Caterina Sarnari, Angela Rizzi and Eleonora Nucera.

[Nutrients] Manuscript ID: nutrients-1705670 - Major Revisions

Author's Reply to the Review Report (Reviewer 2)
Comments and Suggestions for Authors

Thank you for the opportunity to review the manuscript, Epinephrine auto-injector prescription and use: a retrospective analysis and clinical risk assessment of adult patients sensitized to lipid transfer protein. In this manuscript, the authors made use of retrospective data from 165 adults sensitized to LTP, amongst whom there were 68 reactions (predominantly minor/localized symptoms), of which only one required administration of epinephrine. Although an interesting and timely manuscript given an increasing focus on LTP, there are many concerns with the manuscript in its current form.

  1. Both the abstract and the main text lack an explicit aim.
    We thank the Reviewer for the comment. We divided the aims in main and secondary in the abstract and in materials and methods, we changed the order of paragraphs in the text and added a sub-paragraph about aims in Results. We modified accordingly.
  1. In Line 20, the authors report that they investigated “further food reaction” over 3 years. Further to what? As this is the first mention of food reaction related to their study population, it is unclear to what these reactions are further.
    We thank the Reviewer for the comment. We modified accordingly.
  1. Line 21: First aid centres, in English, typically refer to small clinics where first aid for minor injuries is provided. Do the authors mean emergency services?
    We thank the Reviewer for the comment. We modified accordingly.
  1. The abstract lacks a definition of “appropriateness of the epinephrine prescription.” This needs to be operationalized, given that it is one of the primary outcomes.
    We thank the Reviewer for the comment. For the definition of appropriateness of the epinephrine prescription, we referred to EAACI guidelines (ref. Muraro A. al., 2022). We modified accordingly.
  1. What types of statistical analyses were performed? One wonders if the authors performed associations (more correctly) than correlations, although “correlation” is mentioned in Line 29.
    We thank the Reviewer for the comment. We have substituted the word association (Adhering to the statistical procedure used) to the word correlation.
  1. Please clarify what is mean by “pollen platanus”.
    We thank the Reviewer for the comment. We modified accordingly.
  1. In Lines 21-22, the authors refer to “possible predictive factors.” Is this the same as the “cofactor” described in Line 32?
    We thank the Reviewer for the comment. Possible predictive factors were variables such as gender, age, presence of comorbidity. Cofactors were variables such as physical exercise, alcohol consumption. We modified accordingly.
  1. Line 39: “Pan allergens widespread” is redundant. Reword to “allergens widespread”.
    We thank the Reviewer for the comment. We modified accordingly.
  1. Line 40 “carry out” is colloquial. Please revise.
    We thank the Reviewer for the comment. We modified accordingly.
  1. Lines 45-46: “This molecular feature increases…. the occurrence of severe systemic reactions.” Perhaps the authors mean “increases the risk of severe systemic reactions”?
    We thank the Reviewer for the comment. Risk is a term linked to probability, occurrence is linked with absolute frequency of an event. In the article we were dealing with occurrence. We modified accordingly.
  1. Section 2.1: This is more correctly “participants” than “patients”.
    We thank the Reviewer for the comment. We modified accordingly.
  1. Line 85: What do the authors mean by “we programmed to patient an annual visit…”.
    We thank the Reviewer for the comment. We scheduled for each patients an annual visit for a clinical evaluation and the renewal of epinephrine prescription if necessary. We modified accordingly.
  1. In the paragraph from Lines 76-93, the authors write about the appropriateness of a prescription for epinephrine autoinjectors. Yet, there is neither an operationalized definition for appropriateness, nor a discussion as to how appropriateness was determined (who wrote the initial prescription, who determined appropriateness and based on what criteria, etc). Without this information, the reader cannot appropriately interpret these findings.
    We thank the Reviewer for the comment. We evaluated the appropriateness epinephrine prescriptions according to EAACI guidelines. We modified accordingly.
  1. With consideration to the reactions reported, were data collected on the form of the food that triggered the reaction (i.e. cooked, vs raw)?
    We thank the Reviewer for the comment. Considering the LTPs is highly resistant to heat, gastric digestion and any food processing, we did not consider the form of the food culprit ingested by the patients.
  1. Line 113: Rather than qualitative variables, do the authors mean “descriptive” variables?
    We thank the Reviewer for the comment. We modified accordingly.
  1. Line 113: Rather than quantitative variables, do the authors mean “continuous” variables?
    We thank the Reviewer for the comment. Authors meant exactly what they wrote.
  1. In some of the results, observations are less than 5. Rather than a chi-square test, it would be appropriate to use a Fisher Exact Test.
    We thank the Reviewer for the comment. We changed the statistical procedures section accordingly.
  1. The results section is very long. Please consider including subheadings that align with the outcomes of interest to help guide the reader.
    We thank the Reviewer for the comment. We modified accordingly.
  1. Lines 166-7: The loss to follow up is substantial, considering a relatively small sample size. Please comment, and/or consider multiple imputation to maximize the data.
    We thank the Reviewer for the comment. Authors considered the procedure of multiple imputation but, due to observational and retrospective nature of the study, authors preferred not to perform this procedure.
  1. Table 2: A small comment: The degree (superscript o) after the number of each year is not used in English.
    We thank the Reviewer for the comment. We modified accordingly.
  1. Table 2: A small comment: What do the commas mean following some of the results?
    We thank the Reviewer for the comment. We modified accordingly.
  1. In Line 212, the authors mention “logistic regression.” This was a bit of a surprise, given that there was no mention of this type of test in the section describing statistical analysis methods.
    We thank the Reviewer for the comment. We modified accordingly.
  1. Lines 245-7: I suspect that these lines were included in the original template. In the submission, they should have been removed.
    We thank the Reviewer for the comment. We modified accordingly.
  1. Although one of the outcomes of interest was co-factors, it is unclear how these data were collected. This needs to be described in the methods.
    We thank the Reviewer for the comment. We collected data about co-factor implicated in the reactions from patients’ clinical history.
  1. The manuscript lacks a conclusion that aligns with the aim (also missing).
    We thank the Reviewer for the comment. We modified accordingly.

Best Regards

Sara Urbani, Arianna Aruanno, Antonio Gasbarrini, Alessandro Buonomo, Rossana Moroni, Caterina Sarnari, Angela Rizzi and Eleonora Nucera

Round 2

Reviewer 2 Report

The authors have carefully attended to many of my previous comments.

1.       Line 54: The authors use the abbreviation nsLTP, but which is not defined. Based on the associated reference, I presume that this abbreviation refers to non-specific lipid transfer protein. However, this needs to be clarified.

2.       Lines 108-11: As written, this sentence is colloquial. Please revise.

3.       Line 130: Clarify what is meant by “Fisher’s Exact Test when required.” Do the authors mean when observations were 5 or less?

4.       Line 130-132: The authors indicate that data were analysed using logistic regression. Typically, this is reported as an odds ratio (OR) and the 95% confidence interval, including both unadjusted and adjusted models. However, I do not see any results reported as such. Moreover, the authors report a hazard ratio, which is not calculated using logistic regression

5.       Line 134: Provide manufacturer and city for SPSS 25.

6.       Figure 2: The authors refer to “correlations.” Yet in Section 2.4, there is no description of any test of correlation.

7.       When writing a response to reviewer, it is greatly appreciated to copy the text verbatim from the revised manuscript to the response to reviewer. This makes the reviewer’s job much easier.

Author Response

June, 6th 2022

To Guest Editor and Reviewer
Nutrients MDPI

We would like to greatly thank the Editor and Reviewer who encouraged a further revision of the manuscript.

Please find the enclosed the Revision vers. 2 of the Original article entitled “Epinephrine auto-injector prescription and use: a retrospective analysis and clinical risk assessment of adult patients sensitized to lipid transfer protein” by Sara Urbani, Arianna Aruanno, Antonio Gasbarrini, Alessandro Buonomo, Rossana Moroni, Caterina Sarnari, Angela Rizzi and Eleonora Nucera.

[Nutrients] Manuscript ID: nutrients-1705670 - Minor Revisions

Author's Reply to the Review Report (Reviewer 2)
Comments and Suggestions for Authors

Overall
The authors have carefully attended to many of my previous comments.

  1. Line 54: The authors use the abbreviation nsLTP, but which is not defined. Based on the associated reference, I presume that this abbreviation refers to non-specific lipid transfer protein. However, this needs to be clarified.
    We thank the Reviewer for the comment. We modified accordingly in “non-specific lipid transfer protein (nsLTP)”.
  1. Lines 108-11: As written, this sentence is colloquial. Please revise.
    We thank the Reviewer for the comment. We modified accordingly in “Afterward, we conducted a telephonic interview to investigate the reasons of follow-up drop-out, classifying them into: (1) personal issues; (2) logistic reasons (distance from hospital, difficult to schedule an appointment...); (3) poor perception of allergy severity”.
  1. Line 130: Clarify what is meant by “Fisher’s Exact Test when required.” Do the authors mean when observations were 5 or less?
    We thank the Reviewer for the comment. We applied Fisher Exact test when observations were 5 or less.
  2. Line 130-132: The authors indicate that data were analysed using logistic regression. Typically, this is reported as an odds ratio (OR) and the 95% confidence interval, including both unadjusted and adjusted models. However, I do not see any results reported as such. Moreover, the authors report a hazard ratio, which is not calculated using logistic regression.
    We thank the Reviewer for the comment. In fact, we made a mistake writing HR instead of OR. We modified accordingly. Line 225 and 227 were OR and not HR.
  3. Line 134: Provide manufacturer and city for SPSS 25.
    We thank the Reviewer for the comment. We added details into the text.
  1. Figure 2: The authors refer to “correlations.” Yet in Section 2.4, there is no description of any test of correlation.
    We thank the Reviewer for the comment. We modified Figure 2 legend accordingly.

Best Regards

Sara Urbani, Arianna Aruanno, Antonio Gasbarrini, Alessandro Buonomo, Rossana Moroni, Caterina Sarnari, Angela Rizzi and Eleonora Nucera